# A Comparative Review of Tocosomes, Liposomes, and Nanoliposomes as Potent and Novel Nanonutraceutical Delivery Systems for Health and Biomedical Applications

**DOI:** 10.3390/biomedicines12092002

**Published:** 2024-09-03

**Authors:** Omar Atrooz, Elham Kerdari, M. R. Mozafari, Nasim Reihani, Ali Asadi, Sarabanou Torkaman, Mehran Alavi, Elham Taghavi

**Affiliations:** 1Department of Medical Laboratory Sciences, Faculty of Applied Medical Sciences, Al-Ahliyya Amman University, Amman 19328, Jordan; 2Department of Biological Sciences, Mutah University, Mutah 61710, Jordan; 3Australasian Nanoscience and Nanotechnology Initiative (ANNI), Monash University LPO, Clayton 3800, Australia; 4Faculty of Fisheries and Food Science, Universiti Malaysia Terengganu, 21030 Kuala Nerus, Terengganu, Malaysia

**Keywords:** biomedicine, drug delivery, encapsulation, liposome, nanoliposome, nanonutraceutical, tocosome

## Abstract

Contemporary nutraceutical and biomedical sectors are witnessing fast progress in efficient product development due to the advancements in nanoscience and encapsulation technology. Nutraceuticals are generally defined as food substances, or a section thereof, that provide us with health benefits such as disease prevention and therapy. Nutraceutical and biomedical compounds as well as food supplements are a natural approach for attaining therapeutic outcomes with negligible or ideally no adverse effects. Nonetheless, these materials are susceptible to deterioration due to exposure to heat, oxygen, moisture, light, and unfavorable pH values. Tocosomes, or bilayered lyotropic vesicles, are an ideal encapsulation protocol for the food and nutraceutical industries. Biocompatibility, high entrapment capacity, storage stability, improved bioavailability, site specific delivery, and sustained-release characteristics are among the advantages of this nanocarrier. Similar to liposomal carriers and nanoliposomes, tocosomes are able to encapsulate hydrophilic and hydrophobic compounds separately or simultaneously, offering synergistic bioactive delivery. This manuscript describes different aspects of tocosome in parallel to liposome and nanoliposome technologies pertaining to nutraceutical and nanonutraceutical applications. Different properties of these nanocarriers, such as their physicochemical characteristics, preparation approaches, targeting mechanisms, and their applications in the biomedical and nutraceutical industries, are also covered.

## 1. Introduction

The exceptional characteristics of liposomes and allied technologies in the biomedical and nutraceutical sectors have resulted in the manufacture of novel products with improved sensory and health-benefit characteristics [1,2]. Considering the increasing rate of worldwide life-threatening diseases, the usefulness of nano-encapsulated dietary supplements and nutraceuticals for efficacious disease prevention and therapeutics is extremely important. Currently, a major challenge is obtaining a formulation that can exhibit a satisfactory stability and bioavailability of the nutraceutical and nanonutraceutical compounds while maintaining cost-effectiveness. Nanotechnology appears as a promising strategy to augment the stability, improve the solubility, and enhance the permeability of the encapsulated material. The small size of the nanoparticles and their high surface-to-volume ratio, along with the versatility in their composition and number of lamellae, offer great opportunities in formulating novel food and nutraceutical products [3,4]. Nanoparticles and nanovesicles that are being used in the field of nutraceuticals thus far include, but are not restricted to, micelles, nanoemulsions, nanocochelates, nanoliposomes, archaeosomes, and tocosomes [5,6].

Applications of liposomal and nanoliposomal drug carriers in the pharmaceutical sector for drug delivery and targeting have shown promising results and some formulations have already found their way into the market. Consequently, other sectors, including food, biomedical, and nutraceutical industries, have also started to utilize these carriers for the entrapment, encapsulation, and site-specific delivery of numerous bioactive materials, including vitamins, antioxidants, polyphenols, flavor components, omega fatty acids, amino acids, and enzymes [7,8,9]. As the most novel encapsulation system, tocosomes are finding their way in the pharmaceutical and food industries. Tocosomes are potent nanocarriers composed of phosphorylated tocopherols (vitamin E compounds) and can be utilized for the encapsulation and delivery of various bioactive compounds. Tocopherols are known for their exceptional antioxidant properties and biocompatibility. This makes them suitable for developing nanocarriers for the encapsulation of bioactive material, especially in the nutraceutical and pharmaceutical industries. Some of the main advantages of employing tocosomes in the biomedical and nutraceutical industries are depicted in Figure 1. Tocosomes can enhance the therapeutic effects of the encapsulated agents by directly delivering them to the cells. This targeted approach reduces dosages while achieving the desired results, minimizing potential side-effects. Another distinctive advantage of tocosomes is that they can be manufactured by using safe ingredients obtained from natural sources [6].

Due to the afore-mentioned advantages and benefits of nanocarriers, numerous products have been launched into the market using these encapsulation systems [12,13]. Table 1 presents some of the approved nano-encapsulated dietary products that have found their way into the market for human use. Protection of the sensitive bioactive compounds in the gastrointestinal system (and even when present in external environment), storage stability, and enhanced bioavailability in the body are some of the other benefits that are offered by the nanovesicles and nanoparticles for dietary consumption. Accordingly, this article reviews the advantages of tocosomes, liposomes, and nanoliposomes, with emphasis on their physicochemical properties, manufacturing techniques, and targeting mechanisms, as well as their applications in the pharmaceutical, biomedical, food, feed, and nutraceutical industries.

## 2. Main Physicochemical Properties

Nanocarrier techniques are one of the most promising classes of drug delivery and encapsulation systems used in the field of pharmaceutical nanotechnology. These carriers include cochleates, archaeosomes, nanoemulsions, micelles, nanoliposomes, nanostructured lipid carriers (NLCs), and solid lipid nanoparticles (SLNs) [14,15]. When compared to other encapsulation protocols such as metallic or polymeric carriers and niosomes, lipid-based drug carriers possess distinctive advantages. These include the ability to encapsulate (or entrap) molecules with different solubilities, targetability, and the possibility of being manufactured using natural ingredients on the industrial scale. Lipidic vesicles are able to protect the encapsulated compounds against metal ions, free radicals, enzymes, and pH variations that might cause degradation or deterioration of the sensitive material. They provide stability to water-soluble molecules and hydrophilic compounds, particularly in high-water-activity environments. The unique characteristics of lipidic carriers are mainly due to the physicochemical characteristics of their ingredients, i.e., lipids and phospholipids. The hydrophilic sections of the phospholipid molecule are positioned toward the external and internal aqueous phases while the hydrophobic moieties face their counterparts on the other phospholipid molecules to avoid contact with water molecules. The final organization, structure, and physicochemical properties of the lipidic vesicles therefore depend on the type, size, morphology, concentration, and charge of the constituent molecules, as well as the solution properties (e.g., temperature, ionic strength, and pH) [16]. The curvature energy of a bilayer lipid or phospholipid vesicle is higher than in the stacked multilamellar liquid crystalline phase (in the presence of excess water). Therefore, an energy input is required in order to stimulate the curvature of the lipid/phospholipid bilayer into a vesicle structure (i.e., a closed, continuous lipid bilayer) [17]. This means that, unlike what is commonly accepted, the synthesis of liposomes, nanoliposomes, and tocosomes is not a random or spontaneous phenomenon. Curvature of the flat bilayer structure(s) into a spherical vesicle (that can be unilamellar or multilamellar) results in a stable formulation due to the lack of chemical interactions between the hydrophobic moieties and the aqueous medium. Consequently, it can be suggested that the mechanism of the formation of the lipidic vesicles and tocosomes (which also possess lipids and phospholipids in their structure) is the van der Waals forces and hydrophilic–hydrophobic interactions between lipids and/or phospholipids and the water molecules [18].

The amphiphilic and amphipathic nature of phospho/lipid bilayers in aqueous solutions is very similar to the structure of biological cell membranes. This characteristic results in excellent interactions between the lipid vesicles and the mammalian cell membranes and consequently provides effective uptake by cells and tissues [19]. It must be noted that, technically and scientifically, liposomes and nanoliposomes are not identical structures. As a matter of analogy, microparticles and microcapsules are different than their nano-counterparts (i.e., nanoparticles and nanocapsules, respectively) since they have different physical properties. The scale of size dimension of a liposomal vesicle is micrometric while nanoliposomes are within the nanometric size range. Since they are composed of similar ingredients (i.e., lipid and phospholipid molecules), liposomes and nanoliposomes share similar structural and lyotropic attributes. Nevertheless, due to the reduction in size to the nanometric range, nanoliposomes possess a higher surface area (compared to their volume) and, consequently, their drug delivery efficiency increases. Possession of more surface area means that nanoliposomes have distinctive elastic, tensile, and magnetic characteristics. In addition, they possess higher reactivity, increased conductivity, and increased propensity to refract and reflect light [20].

Lipid vesicles vary in charge (zeta potential), number of lamellae (bilayers), and in size depending partially on the method of synthesis (see the next section) and the characteristics of their ingredients. With respect to the number of bilayers, lipid vesicles are classified into several groups, designated as unilamellar, oligolamellar, or multilamellar [4,16]. The size range of small unilamellar vesicles (SUVs) is usually between 20 and 100 nanometers, double-bilayer vesicles (DBVs) are in the size range of 200–500 nm, oligolamellar vesicles (OLVs) are in the size range of 500–700 nm, the size range of multilamellar vesicles (MLVs) is roughly between 0.5 and several micrometers, the large unilamellar vesicle (LUV) size range varies from 0.1 to 1.0 μm, giant unilamellar vesicles (GUVs) are in the size range of 2.0 μm and above, and multivesicular vesicles (MVVs) can be in the size range of 5.0 μm and larger (Table 2) [20,21,22].

Tocosomes possess a number of important advantages when compared with other dispersed systems, including a high encapsulation of water-soluble and lipid-soluble compounds, cost effectiveness, and reproducible sustained release rates [6]. Tocosomal formulations can replace some commercially available products containing toxic solubilizing agents. Therefore, they provide effective and safe alternative dosage forms for parenteral, intravenous, transdermal, and oral administration [21,22]. Similarly, these vesicles can be used in the formulation of novel nutraceutical, food, and feed products since they provide improved ingredient stability, enhanced bioavailability, and controlled release of the encapsulated molecules. Furthermore, they can be utilized in the development of functional and clean label food products, ultimately meeting consumer demands for healthier and more innovative food options.

## 3. Preparation Methods

The main physicochemical characteristics of vesicular and colloidal drug carriers, including tocosomes, predominantly depend on the type, concentration, and ratio of ingredients, solvents, and co-solvents used in their manufacture. These attributes include the mean hydrodynamic size, polydispersity index (PDI), zeta potential (i.e., surface charge), number of bilayers, stability, permeability, surface activity, and safety of the vesicles. The amphiphilic bilayers of liposomes and nanoliposomes are mainly composed of lipid and/or phospholipid molecules, while tocosomes are composed of tocopherol phosphate (TP) and di-tocopherol phosphate (T_2_P), along with lipids/phospholipids [18,23]. However, there are other crucial components, without which the construction of a successful tocosomal formulation would be impossible (Figure 2). Furthermore, there are certain considerations for the effective synthesis of these vesicles for food, feed, and nanonutraceutical applications, as listed below.

In order to ensure optimal encapsulation efficiency and stability, high-quality TP, T_2_P, phospholipids, excipients, and bioactive compounds must be employed;To achieve desired characteristics and performance, formulation parameters (e.g., TP, T_2_P, lipid-to-drug ratio, ingredients’ composition) must be optimized;Appropriate sterilization and aseptic techniques must be applied if tocosomes are intended for human or animal use;The regulatory guidelines and standards for the manufacturing and labeling of nutraceutical products must be followed [4,8,24].

As depicted in Figure 2, the essential components of a tocosome vesicle are (i) tocopherol phosphate (TP); (ii) di-tocopherol phosphate (T_2_P); (iii) lipid/phospholipid; (iv) one or more antioxidant; (v) helper lipid (can function as a charge-inducer or stabilizing agent); and (vi) sterol. The most generally used phospholipid ingredient for the construction of liposomes and other lipid-based carriers is lecithin (phosphatidylcholine). This readily available ingredient can also be used for the manufacture of tocosomes. Due to its amphipathic nature, lecithin is immiscible with water molecules and can be isolated from soy or egg yolk economically. However, it is also present in liver, peanuts, and wheat germ [24]. The characteristics and ratio of the phospholipid ingredients and the synthesis method of vesicles determine if a single or multiple bilayers will be formed. Fatty acids can also be used as a component of lipid vesicles, and their degree of saturation (i.e., the number of double bonds present in their hydrocarbon chain) depends on their source. Animal sources contain more saturated fatty acids compared to plant sources. These ingredients affect the phase transition temperature (Tc), that is, the conversion from a gel (or solid) state to the leakier liquid crystalline form. Large polar molecules and different sugars cannot penetrate through the bilayers of liposomes, nanoliposomes, or tocosomes. However, small lipophilic molecules can penetrate through the bilayered membranes if they are dispersed in the suspension medium. Potassium ions, hydroxyl ions, proteins, and peptide molecules can pass through the bilayers very slowly [18,25].

For each phospholipid molecule in particular, and for surfactants in general, there is a specific concentration at which these molecules assemble in the form of a spherical vesicle when placed in an aqueous medium. This specific concentration is known as the “critical micelle concentration” (CMC). It should be noted that the CMC value depends on the structure and charge of the hydrophilic head group and the structure and the degree of saturation of the hydrophobic tail group of the amphiphilic molecules (e.g., phospholipids). Since, thermodynamically, the sphere shape is the most stable geometric structure (at the lowest energy state), the first micelle formed will be spherical. Along with the assembly of the tail groups of phospholipids together, the entropy of the system decreases. This assembly, which happens principally due to the hydrophobicity of the phospholipid tail groups, eliminates their undesirable contact with the water molecules in which they are suspended. As a result of the combined effect of the mentioned factors, stable spherical vesicles will be formed. At concentrations below the CMC value, the adsorption of the phospholipid ingredients at the interface between air and water will depend on their concentration. Therefore, increasing the concentration results in an increase in adsorption and hence a decline in surface tension. Nevertheless, after attaining the CMC level, the additional amphipathic molecules, which are added to the bulk phase, will simply form micellar structures in an energy-efficient manner. This is due to the fact that the surface tension of the liquid phase will remain constant after reaching the CMC point and does not vary with the concentration changes [26]. Compared to micelles (which are composed of one single layer of molecules), liposomes, nanoliposomes, and tocosomes [6] have the advantage of being composed of bilayers that are more stable than monolayers and provide more protection to the encapsulated compounds. The CMC of most commonly used phospholipids is in the nanomolar range and the concentration of phospholipids used for liposome/nanoliposome manufacturing is much above the CMC. This fact, along with the three-dimensional shape of each phospholipid, leads to the formation of bilayered vesicles when the ingredients are exposed to an aqueous medium [27,28].

There are numerous techniques that are currently available for the manufacturing of drug delivery vesicles and nanocarrier systems. Questions to be addressed right at the beginning of method selection include whether we intend to produce a small quantity of vesicles (i.e., laboratory scale) or if our aim is large-scale manufacture. Another point to be seriously considered is the employment of potentially toxic/volatile solvents (e.g., methanol, chloroform, etc.) and harsh procedures (such as sonication, high pressure homogenization, microfluidization, etc.) or selection of safe (green) techniques. While the majority of conventional liposome preparation techniques require the utilization of toxic solvents and/or deleterious procedures, fortunately, the number of green procedures (e.g., the Mozafari method) is increasing [29,30]. Considering the physicochemical similarities between the ingredients of lipid vesicles and tocosomes along with their microscopically similar colloidal/bilayer structures, procedures used for the manufacture of lipid vesicles can be readily adapted for the synthesis of tocosomes.

Among the conventional techniques, two procedures are commonly used to produce liposomal and nanoliposomal formulations for food and pharmaceutical applications. The first procedure is hydration of the ingredients followed by high-shear force mixing using a sonication technique (by employing bath or probe-type sonicators) or high-pressure homogenization. The size variation of the vesicles is then uniformed by employing extrusion or filtration. In the second technique, phospholipids and other ingredients are first dissolved in an organic solvent (usually chloroform, methanol, ethanol, or a combination thereof) and then placed in an aqueous medium with vigorous agitation under an inert atmosphere. The organic solvents are then evaporated under reduced pressure. The size of the resulting vesicles can be reduced and homogenized by filtration or extrusion techniques. Generally, the first procedure results in vesicles with multiple lamellae (i.e., several bilayers), while the second technique results in vesicles with a lower number of lamellae [19,31,32].

The sonication technique is a relatively simple procedure for decreasing the average size of liposomes and obtaining nanoliposomes. The usual laboratory procedure involves treating the hydrated liposomes with a titanium-tipped probe sonicator in a temperature-controlled setting for several minutes. Probe-tip sonicators generate high-energy input in the lipid suspension; however, they suffer from overheating the mixture and hence causing degradation of the molecules. Moreover, sonicator tips discharge titanium particles into the product, which must be removed by centrifugation or sedimentation prior to use. For these reasons, bath-type sonicators are the most widely applied equipment for the manufacture of nanovesicles [33].

Thanks to the advancements in the scientific fields of drug delivery and encapsulation technology, preparing vesicles without employing any toxic solvent or detergent and without employing harsh treatments has become possible. Examples of such green techniques include the polyol dilution method [34], the bubble method [35], the heating method [36], and the Mozafari method [37,38]. As explained above, a sufficient quantity of energy must be provided to induce the curvature of the phospholipid bilayer sheets in order to form stable spherical structures. Although the spherical shape is at the minimum thermodynamic energy level, for the formation of spherical vesicles, the system has to be given a minimum level of energy, known as “the activation energy.” The form of the required energy can be either mechanical, physical, acoustic (e.g., sonication), thermal, or a combination of them. In general, the synthesis methods of lipidic vesicles are classified as low-energy and high-energy protocols. Low-energy methods include solvent diffusion, ethanol injection, and the Mozafari method. On the other hand, high-energy methods include homogenization, sonication, and high-pressure microfluidization procedures [39].

In both of the heating and Mozafari methods developed in our laboratory, two types of energy are applied, namely, mechanical energy (mild agitation) and thermal energy. Although the level of thermal energy required by the heating method is high, in the Mozafari method, the maximum temperature required does not exceed 70 °C [33,38,39]. To ensure the manufacture of successful drug delivery formulations, the following points must be taken into consideration:(1)Biological and physicochemical properties of the material to be encapsulated.(2)Ideal level of encapsulation or entrapment efficiency.(3)The route of administration of drug or other bioactive molecules/compounds.(4)The shelf life and stability of the end product.(5)Characteristics of the medium or solvent(s) in which the lipid vesicles, drug molecules, and other excipients of the formulation are dispersed.(6)Required size, charge, polydispersity index, and release profile of the formulations.(7)Safety profile, potential toxicity, and effective concentration of the encapsulated material in the vesicles.(8)Number of steps and vessels required in the manufacturing protocol.(9)Industrial scalability of the procedure, particularly in order to ensure both constant quality (regarding batch-to-batch variations) and effectual manufacture yield [36,37,38,39].

The ideal manufacturing technique must address all of the afore-mentioned parameters sufficiently. Among these parameters, particle size and polydispersity index are very important factors that influence the bioavailability, safety, and targetability of the carrier formulations [40,41].

## 4. Targeting Strategies

Currently, a main challenge in drug encapsulation and delivery is how to diminish or completely eliminate adverse side-effects of the therapeutic compounds. If bioactive compounds act merely at their target site to produce the required effect without causing impairment to other systems, their effectiveness will enhance significantly. Thanks to their exceptional properties, lipid-based vesicles and tocosomal drug carriers improve the performance of the encapsulated material by rectifying solubility problems, improving bioavailability, increasing stability, and establishing adequate drug concentrations at the target cells and sub-cellular compartments in order to attain optimum disease prevention and/or therapy [42].

Application of tocosomal delivery systems in drug targeting offers many advantages when compared to direct conjugation of a targeting moiety to the drug. Firstly, availability of the functional ligands for direct conjugation to a therapeutic agent may be limited, causing the conjugation chemistry to become challenging. Secondly, the biological activity of the medicament can be affected, requiring the additional formation of a cleavable linker to enable drug release after endocytosis. Furthermore, multiple bioactive agents can be delivered to their target upon encapsulation by a single vesicle. This is while generally only one type of drug is delivered after the uptake of the directly conjugated therapeutic formulations. Consequently, targetable tocosomal systems are preferred over directly conjugated medicaments. The targeted delivery of tocosomal formulations can be attained by either passive or active targeting mechanisms as explained in detail in the following sections [43].

The active targeting mechanism involves the direct movement of the drug carriers to their target tissue, organ, or cell before releasing the encapsulated or entrapped material. It should be noted that while “encapsulation” refers exclusively to the localization of drug molecules in the internal aqueous phase of the tocosomal vesicles, “entrapment” can refer to the presence of drug molecules in the lipid phase (bilayers) of the vesicles. Active drug targeting can be achieved by implementing appropriate modifications to the structure of the drug carriers. For active drug targeting, different specialized liposomes and nanoliposomes have been designed and formulated so far. These include thermo-labile, photo-sensitive, pH-sensitive, antibody-coated vesicles, and magnetic vesicles. Active drug targeting can also be accomplished by employing external stimuli, including ultrasound, laser, or magnetic fields, as well as different sources of light energy [44,45].

In passive targeting, the drug–carrier complex arrives to its destination by following the natural physio-anatomical conditions of the body. Site-specific delivery is achieved based on the physicochemical characteristics of the carrier system and does not necessitate application of any targeting protocol. The in vivo biodistribution and clearance kinetics of the vesicles depend on certain physicochemical parameters, such as particle size, polydispersity index (PDI), zeta potential, and hydrophobicity. These factors can be manipulated in order to achieve passive drug targeting [40]. Vesicles with an average 100 nm diameter, for instance, can extravasate selectively in the body tissues possessing leaky vasculature, such as solid tumors, while vesicles with a larger particle size (up to 1 μm) are taken up by the reticuloendothelial system (RES) passively. Stealth vesicles are one type of lipid vesicle that can be manufactured by covering the surface of the vesicles with hydrophilic chains (e.g., polyethylene glycol) in order to prevent opsonization. This approach will provide the vesicles with a long circulation time, less elimination from the blood stream, and, consequently, higher drug uptake. Nanocarriers will move to and accumulate in the tumorous or infected tissues or organs, and, as the stealth vesicles become degraded, they will release their drugs at the target area in the body. This is an example of a passive drug targeting strategy due to the fact that the stealth vesicles move to their target and treat the affected tissues or organs passively, without incorporating any active targeting mechanism [46,47].

One example of an in vitro targeting mechanism in food systems is the employment of the liposomal lysozyme formulation, which has the tendency to target the regions in the cheese matrix where the bacteria are accumulated [48]. Another instance of nanoliposome targeting in food products is the utilization of the nanocarriers encapsulating the antibacterial agent nisin (manufactured by the Mozafari method) in order to target and eliminate food contaminants [49]. On the other hand, in vivo targeting can be exemplified by considering the targeted delivery of probiotic compounds in the gut system. A crucial point in the utilization of probiotics is that, for optimal functionality and maximum health benefits, probiotics need to be delivered in specific areas of the digestive system. Although the digestive tract is a highly organized system, it is also very heterogenic and multicompartmental. For instance, the immunological system of the digestive tract is unique in every single section. Consequently, to exploit the immune-regulatory effect of probiotics, they should be targeted at and released in the area where immunological signaling occurs. Nonetheless, this takes place in the ileum, where Peyer patches are present, or more distal, where other gut-associated lymphoid structures are present. Targeting the probiotics at the intended location inside the digestive tract and their controlled release can be achieved by using certain nanocarrier systems, such as tocosomes and nanoliposomes (Figure 3) [4,6,48].

## 5. Applications in Food and Nutraceutical Industries

The encapsulation of nutraceutical compounds refers to the process of enclosing these bioactive substances within a protective matrix or shell. Nanonutraceuticals are substances that provide health benefits beyond basic nutrition, such as vitamins, minerals, antioxidants, and other dietary supplements (see Table 3) [48,49,50,51]. Encapsulation offers several advantages for the delivery and enhanced efficacy of nutraceutical compounds:(1)Protection: Encapsulation shields the sensitive nutraceutical compounds from degradation due to light, oxygen, moisture, and other environmental factors. This protection helps to maintain the stability and potency of the compounds during storage and transportation.(2)Controlled release: Encapsulation allows for the controlled release of nutraceuticals in the body. By regulating the rate of release, encapsulation can optimize the absorption and bioavailability of these compounds, ensuring maximum effectiveness.(3)Masking taste and odor: Some nutraceutical compounds may have an unpleasant taste or odor. Encapsulation can mask these sensory properties, making the products more palatable and consumer-friendly.(4)Enhanced solubility: Encapsulation techniques can improve the solubility of poorly soluble nutraceutical compounds, enhancing their absorption and bioavailability in the body.(5)Targeted delivery: Encapsulation can facilitate targeted nutraceutical delivery to particular sites inside the body, such as the gastrointestinal tract or bloodstream. This targeted approach can enhance the therapeutic effects of the compounds while minimizing side-effects.(6)Combination therapy: Encapsulation enables the combination of multiple nutraceutical compounds into a single dosage form, allowing for synergistic effects and simplified administration for consumers [4,48,50,51].

There are various encapsulation techniques employed in the food and nutraceutical industries, including spray drying, coacervation, emulsification, nanoparticle formation, tocosomes, liposomes, and nanoliposomes, among others. Selection of the encapsulation technique depends on factors such as the physicochemical properties of the nutraceutical compounds, desired release profile, and intended application of the final product [37,51,52]. Application of the nanovesicles as potential carriers to encapsulate and deliver food, feed, and nutraceutical compounds is a rapidly growing field. The recent invention of the Mozafari method and pro-liposome techniques offer possible solutions to many of the food processing problems. Furthermore, continuing research into the application of cost-effective commercial phospholipids and other tocosomal ingredients will lead to suitably priced products [49,53].

Studies so far suggest the potential of nanocarrier systems for improving the flavor of ripened cheese using accelerated methods, the targeted delivery of functional food ingredients, the synergistic delivery of tocopherols and ascorbic acid for enhancing antioxidant activities in foods, and the stabilization of minerals (such as calcium, zinc, and iron) in milk and other drinks. In the food, feed, and nutraceutical industries, liposomes and other lipidic carriers have been employed to encapsulate flavoring and dietary compounds [54,55]. They are also suitable candidates for entrapping and delivering antimicrobial preservatives in order to improve product shelf life [52,56]. A number of scientific publications describe the employment of nanocarrier systems to modify the pharmacokinetic characteristics of drugs, vitamins, herbs, and even enzymes (for a recent comprehensive review, see Ref. [39]). Moreover, some colloidal vesicles have been employed for targeted delivery of the encapsulated compounds in dairy products. The main research and development thus far have aimed to change the texture of food components, develop new tastes and sensations, accelerate cheese ripening, regulate the release of flavors, and increase the absorption and bioavailability of nutritional compounds. In the dairy industry, the first application of lipid-based vesicles was in the cheese-ripening process [57]. Furthermore, different types of lipidic nanocarriers, such as nanoliposomes, have been used for the entrapment or encapsulation of antimicrobials, enzymes, and minerals (e.g., calcium, zinc, and iron) in different dairy products [57,58]. The targeted delivery of dietary supplements and nutraceuticals, using lipid vesicles, results in an enhanced stability and bioavailability of the formulation. Some of the main classes of nutraceutical compounds considered for encapsulation by tocosomes are antioxidants [59,60], minerals [61], bioactive peptides [62], polyunsaturated fatty acids [63], dye and flavoring [64,65], vitamins [66], and enzymes [67,68].

## 6. Biomedical Applications

As mentioned before, vesicular lipid-based drug carrier systems as well as tocosomes can accommodate both water-soluble and lipid-soluble ingredients separately or simultaneously (e.g., when a synergistic effect is required) in addition to amphiphilic compounds [69,70,71]. These bioactive carriers are biocompatible and biodegradable and are able to provide long-lasting and controlled drug release. Their unique properties can positively affect the pharmacokinetics and biodistribution of the encapsulated material. Tocosomes are also an ideal technology for enzyme replacement therapy and can be utilized in antiviral, antifungal, and tumor prevention and cure. The main ingredients of tocosomes (i.e., TP and T_2_P) are very unique molecules with exceptional health benefits and medicinal properties. The phosphorylated form of alpha-tocopherol is known as alpha-tocopherol phosphate (TP) and is present naturally in human and some animal tissues, as well as in certain fruit and vegetables [72,73]. Clinical studies thus far have revealed that TP and T_2_P molecules have several health benefits and medicinal properties, including anti-inflammatory, atherosclerotic-preventing, and heart-protective properties. Furthermore, the repressive effect of TP molecules toward tumor invasion has also been documented [39,74,75]. Research has also revealed that TP and T_2_P molecules have protected primary cortical neuronal cells from glutamate-induced cytotoxicity in vitro. They have also reduced the levels of lipid peroxidation moieties in the liver and blood plasma of rodents [76].

Based on the above-mentioned proven advantages, tocosomes can be employed efficiently as carriers for small molecules (e.g., vitamins, minerals, or chemotherapeutic drugs) and for the encapsulation of large molecules such as cytokines, as well as genetic material [3,6,77]. Tocosomal technology can also be used in radiopharmaceuticals, immunological products, cosmetics, cosmeceuticals, and dermatological formulations. Moreover, they can be employed for enzyme encapsulation and immobilization. They have unique emulsifying properties, can be used to stabilize emulsions, and are good wetting agents. Therefore, they can coat the surface of crystals to make them hydrophilic, as is the case with liposomal and nanoliposomal vesicles [78,79].

Bilayered vesicles, both lipidic and tocosomal, have great potential for applications in the field of genetic engineering as gene and oligonucleotide delivery [80], in biological research as models of cell and biomembranes, and in the synthesis of different vaccines [81,82]. When the drug and other bioactive materials are encapsulated in the tocosomal delivery systems, they are protected against enzymes and other degradation factors in the body. Moreover, consumers will be protected against the harmful side-effects of the encapsulated medications. In the case of controlled delivery or prolonged release, drug discharge from the vesicles depends on the ingredients of the bilayers, as well as bilayer permeability and the physicochemical characteristics of the encapsulated bioactive material [83]. Bioactive release can also take place as a result of phase variations of the bilayer ingredients in response to external stimuli, including variations in the temperature or pH values. These external stimuli can destabilize the bilayers, affecting the phase transition temperature of tocosomal components, and result in drug release. Tocosomal and liposomal drug delivery systems have been successfully employed for targeting their load to specific cells in vitro and in vivo as mentioned in Section 4. An example of using tocosomes and nanoliposomes for cancer targeting is depicted in Figure 4 [84].

## 7. Summary and Perspectives

Based on the proven pharmaceutical and biomedical success of lipid vesicles, the tocosomal drug delivery system has recently been introduced to the scientific community. Compared to other drug delivery systems, liposomes and nanoliposomes are the most applied carriers for the encapsulation and sustained release of hydrophilic, hydrophobic, and amphiphilic material due to their superior safety and biocompatibility profiles. These lipidic vesicles can protect different types of bioactive agents from deterioration and degradation. As a closely related encapsulation technology, tocosomes can also provide targetability and improve the bioactivity, solubility, and bioavailability of the encapsulated material. They minimize the adverse effects of different molecules and compounds on the integrity and sensory properties of food products. Furthermore, these carrier systems can be manufactured using natural and economical sources of ingredient molecules, including sunflower, soy, egg yolk, and different nuts and seeds. This attribute makes them suitable for large-scale industrial applications in the biomedical field. Being a new technology, introduced in 2017, tocosomes have not yet found their way into the pharmaceutical market. This can only be achieved after successful outcomes of the preclinical and clinical phases followed by approval by the FDA. The main challenges facing their commercial use, however, are their relatively low physical stability, high sensitivity to pH variations and temperature, and untimely release of the encapsulated material upon long-term storage. Therefore, investigations are being carried out on the use of ingredients, which can modify and stabilize vesicle structures and also improve the production methods. Some of the strategies for improving the stability of tocosomes are the surface coating of the vesicles with certain biopolymers, application of cationic or anionic moieties, changes in the composition of the bilayers, application of membrane stabilizing agents such as alpha- or gamma-tocopherol, sterols, or glycerol, and storage of the vesicles at low temperatures and oxygen-free atmospheres, as well as lyophilization. Nevertheless, more research and optimization need to be carried out to increase the efficiency of the vesicles under in vivo conditions, optimization of their stability, and their commercialization. Furthermore, it is necessary to manufacture the tocosomal vesicles without application of potentially toxic solvents, detergents, and high-shear-force procedures by economical and scalable techniques to achieve suitable formulations for pharmaceutical, nutraceutical, and biomedical applications.

## Figures and Tables

**Figure 1 biomedicines-12-02002-f001:**
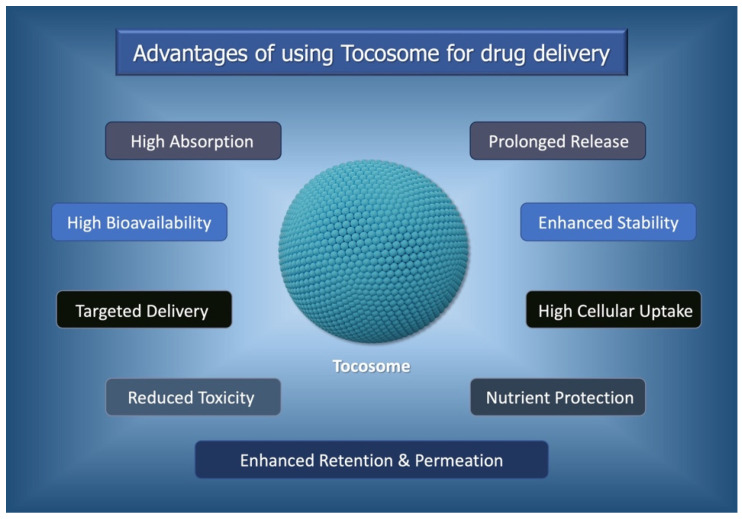
Some of the main advantages and benefits of using tocosomes in the biomedical and nutraceutical industries. Details about the mentioned advantages can be found in References [7,8,9,10,11].

**Figure 2 biomedicines-12-02002-f002:**
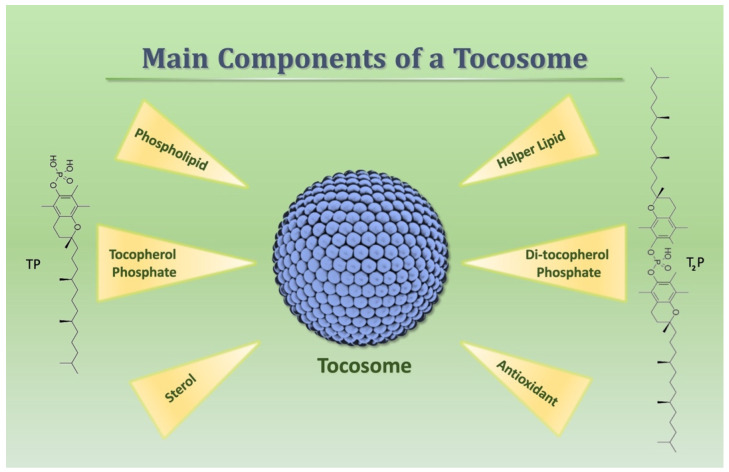
Schematic representation of main ingredients of tocosomes. Main ingredients are the phosphorylated derivates of alpha-tocopherol. Phospholipids can be selected from natural sources or synthetic molecules. A generally employed antioxidant in the structure of tocosomes is vitamin E. A secondary lipid molecule, in a low concentration, can be used as a helper lipid. Sterols used in the formulation of tocosomes mostly include cholesterol, while phytosterols are also utilized.

**Figure 3 biomedicines-12-02002-f003:**
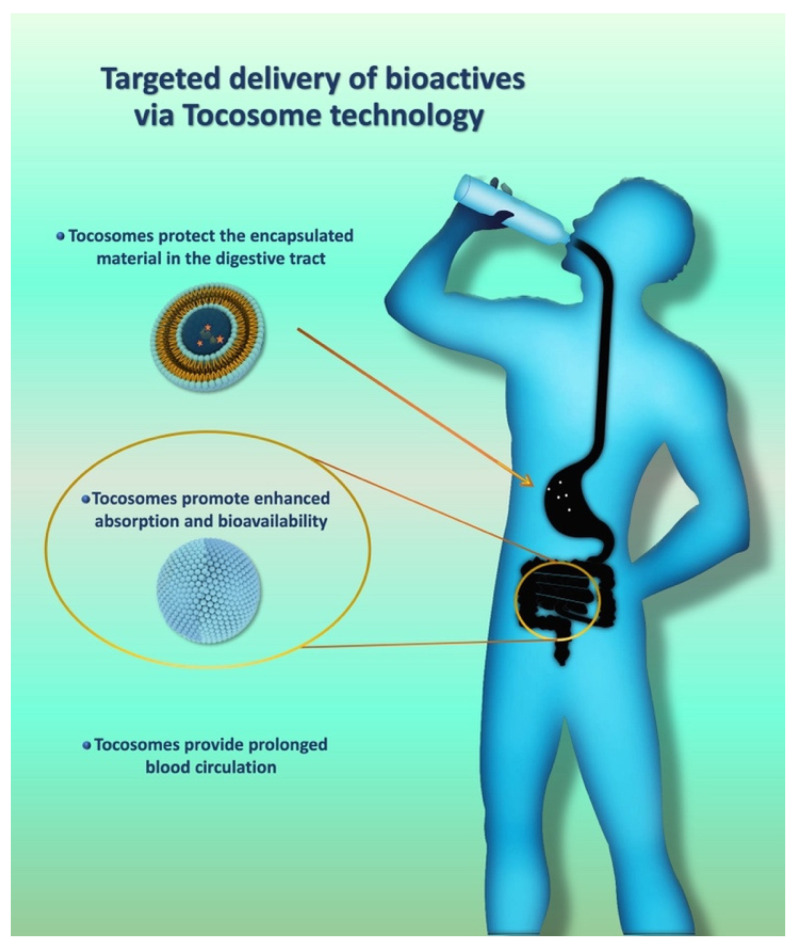
Advantages and benefits of drug targeting in the digestive system using tocosomes.

**Figure 4 biomedicines-12-02002-f004:**
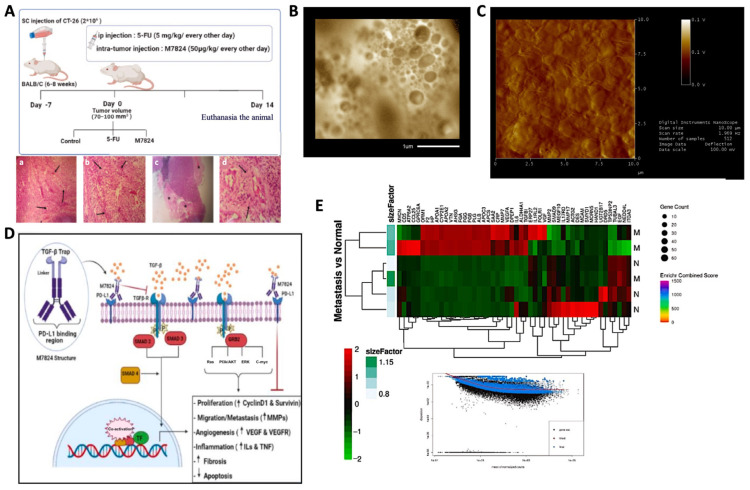
Promising applications of tocosomes and nanoliposomes to combat cancer. (**A**) Experimental design of an in vivo study from our group (showing tumor induction in mice). Tumor tissues revealed more vascular density and (**a**) RBC extravasation; (**b**) inflammation; (**c**) necrosis; and (**d**) tumor stroma (magnifications 10×). (**B**) Confocal laser microscopy of tocosomes. (**C**) Atomic force microscopy of lipidic nanovesicles. (**D**) Schematic presentation of the molecular mechanisms of M7824 (a class of bifunctional fusion proteins) on TGF-b and PD-L1 pathways. (**E**) Hierarchically clustered cancer-related and normal genes. Partially taken from data presented/published in [6,84] with permission (copyright Elsevier).

**Table 1 biomedicines-12-02002-t001:** Licensed dietary products in the market using encapsulation technologies (websites accessed on 23 July 2024).

No.	Product	Manufacturer	Webpage
1	Liposomal Vitamin K_2_ + D_3_	California Gold Nutrition	www.californiagoldnutrition.com
2	Nanoliposomal Vitamin C	NOW Foods	www.nowfoods.com
3	Vesicular Phospholipid Complex	Body Bio	https://bodybio.com
4	Liposomal Creatine	Codeage	www.codeage.com
5	Liposomal Glutathione	Codeage	www.codeage.com
6	Liposomal Carnosine	Dr. Mercola	www.mercolamarket.com
7	Pro-Liposomal Apigenin	MCS Formulas	www.mcsformulas.com
8	Liposomal Curcumin	Lipolife	https://lipolife.co.uk
9	Liposomal Zinc	Vinco	https://vinco.com.au
10	Liposomal Resveratrol	Renue By Science	https://renuebyscience.com
11	Liposomal Vitamin C + Zinc	PlantaCorp	https://plantacorp.com
12	Liposomal Quercetin	PlantaCorp	https://plantacorp.com

**Table 2 biomedicines-12-02002-t002:** Classification of vesicular carriers including ‘tocosomes’ (from References [20,21,22]).

Item	Vesicle Type	Abbreviation	Number of Lamellae	Size Range *
1	Small unilamellar vesicle	SUV	One bilayer	20–100 nm
2	Double-bilayer vesicle	DBV	Two bilayers	200–500 nm
3	Oligolamellar vesicle	OLV	Between two to several bilayers	500–700 nm
4	Large unilamellar vesicle	LUV	One bilayer	0.1 to 1.0 μm
5	Multilamellar vesicle	MLV	More than 10	0.5 to several μm
6	Large multilamellar vesicle	LMV	More than 10	1.0 to several μm
7	Giant unilamellar vesicle	GUV	One bilayer	>2.0 μm
8	Multivesicular vesicle	MVV	Several vesicles encapsulated inside a giant vesicle with one bilayer	>5.0 μm

* Approximate values.

**Table 3 biomedicines-12-02002-t003:** Characteristics of some of the commonly used nutraceuticals and health-benefit compounds considered for encapsulation using tocosomes and other vesicular drug delivery protocols (from References 4, 7, 25, and 50 with amendments and improvements).

Item	Bioactive Compound	Health Benefits	Source	Molecular Weight (g/mol)
1	Astaxanthin	Protects against UV damage, boosts immune system, reduces inflammation	Algae, yeast, shrimp, and different fish	596.841
2	Berberine	Treatment of different types of cancer	Barks, twigs, leaves, stems, roots, and rhizomes of various medicinal plant species	336.3612
3	Curcumin	Antibacterial, antioxidant, and anti-inflammatory effects	Turmeric	368.38
4	Hesperidin	Therapeutic potential in heart disease, blood vessel disorders, metabolic disorders, and neurodegenerative diseases	Citrus fruits	610.1898
5	Kaempferol	Antioxidant and anticarcinogenic effects	Apples, peaches, tomatoes, grapes, green tea, lettuce, broccoli	286.23
6	Luteolin	Anticarcinogenic effects	Green pepper, oregano, carrots, broccoli	286.24
7	Quercetin	Promotes cardiovascular health and improves blood flow	Grapes, onions, lemon tea, citrus fruits, etc.	302.236
8	Resveratrol	Protects against Alzheimer’s disease, cardiovascular diseases, cancer, liver ailments, obesity, diabetes, etc.	The skin of peanuts, grapes, raspberries, mulberries, and blueberries	228.25
9	Rutin	Treats conditions associated with poor blood flow, chronic pain, and high cholesterol	Buckwheat, Japanese pagoda tree, and Eucalyptus	610.517
10	Tocopherol Phosphates *	Anti-inflammatory, atherosclerotic-preventing, and cardioprotective effects	Green vegetables, dairy products, cereals, different seeds and nuts	TP: 510.69 T_2_P: 923.54
11	Omega-3 fatty acids	Can fight inflammation and fight autoimmune diseases, promotes brain health during pregnancy, can fight depression and anxiety	Fish oil, flaxseed, krill oil, squid oil	909.39
12	Ubiquinone (Coenzyme Q10)	Possesses health benefits for cardiovascular diseases, energy-boosting, reduces the symptoms of mitochondrial disorders	Oily fish (such as salmon and tuna), organ meats (such as liver), and whole grains	863.34

* Two forms of tocopherol phosphates are used as main ingredients of tocosomes, i.e., tocopheryl phosphate (TP) and di-tocopheryl phosphate (T_2_P).

## Data Availability

Authors declare no new data were created for this review article.

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
