# Peer review of "A Comparative Review of Tocosomes, Liposomes, and Nanoliposomes as Potent and Novel Nanonutraceutical Delivery Systems for Health and Biomedical Applications"

_biomedicines, 2024, doi:10.3390/biomedicines12092002_

Round 1

Reviewer 1 Report

Comments and Suggestions for Authors

biomedicines-3150739

Tocosomes as potent and novel nanonutraceutical delivery systems for health and biomedical applications

The manuscript by Atrooz et al. summarized the properties and applications of liposomes and tocosomes in biomedical sciences. The authors provided a comprehensive overview of the proposed topic. The manuscript was informative and can be considered for publication. However, there are several issues in this manuscript, as follows.

1. The focus of this review, as stated in lines 77 – 81, was “tocosomes, liposomes and nanoliposomes, with emphasis on their physicochemical properties, manufacturing techniques, targeting mechanisms, as well as their applications in the pharmaceutical, biomedical, food, feed and nutraceutical industries”. The title should be revised appropriately.

2. Figures 1 and 4 are meaningless and unnecessary.

3. Section 3: The authors should separate the content into sub-sections. For each method, please mention more details as well as the advantages and limitations. Finally, it is best to summarize those methods into a table with brief information for comparison.

4. Are there any tocosome products on the market? If not, please discuss the challenges of the tocosome development and how to overcome them.

5. Please correct typos and punctuation errors (such as in lines 80, 175, etc.)

Comments on the Quality of English Language

Some typos and punctuation errors should be corrected.

Author Response

Comments 1: [The focus of this review, as stated in lines 77 – 81, was “tocosomes, liposomes and nanoliposomes, with emphasis on their physicochemical properties, manufacturing techniques, targeting mechanisms, as well as their applications in the pharmaceutical, biomedical, food, feed and nutraceutical industries”. The title should be revised appropriately.]

Response 1: [We thank the reviewer for this instructive comment. We have accordingly changed the title.]

Comments 2: [Figures 1 and 4 are meaningless and unnecessary.]

Response 2: [Authors would like to thank the reviewer for insightful suggestion. We have removed Figure 4 completely. However, Figure 1 could not be removed due to the comment and suggestions of the other Reviewer, based on which new References are now added to the Figure legend in support of the claims in the image.]

Comments 3: [Section 3: The authors should separate the content into sub-sections. For each method, please mention more details as well as the advantages and limitations. Finally, it is best to summarize those methods into a table with brief information for comparison.]

Response 3: [Authors appreciate the constructive suggestion of the respectful Reviewer. Section 3 is in fact a summarised and highly rephrased version  of our recently published comprehensive review articles listed below. In order to avoid plagiarism and repetition we would humbly request the section to remain as it is. The mentioned recent publications are:

1. Jalilian Z, Mozafari MR, Aminnezhad S, Taghavi E. Insight into heating method and Mozafari method as green processing techniques for the synthesis of micro-and nano-drug carriers. Green Processing and Synthesis. 2024 Feb 12;13(1):20230136.

2. Alavi M, Rai M, Varma RS, Hamidi M, Mozafari MR. Conventional and novel methods for the preparation of micro and nanoliposomes. Micro Nano Bio Aspects. 2022 May 1;1(1):18-29.

3. Maleki G, Bahrami Z, Woltering EJ, Khorasani S, Mozafari MR. A review of patents on" mozafari method" as a green technology for manufacturing bioactive carriers. Biointerface Research in Applied Chemistry. 2023;13(1).

4. Alavi M, Mozafari MR, Hamblin MR, Hamidi M, Hajimolaali M, Katouzian I. Industrial-scale methods for the manufacture of liposomes and nanoliposomes: pharmaceutical, cosmetic, and nutraceutical aspects. Micro Nano Bio Aspects. 2022 Oct 23;1(2):26-35.

and more ...]

Comments 4: [4. Are there any tocosome products on the market? If not, please discuss the challenges of the tocosome development and how to overcome them.]

Response 4: [There is no tocosome product on the market as yet. This is because they have only been introduced in 2017 and have not yet received FDA approval. We have added new text to the final section of the article in comply with the suggestion of the respectful Reviewer.]

Comments 5: [Please correct typos and punctuation errors (such as in lines 80, 175, etc.).]

Response 5: [In comply with the comment of the Reviewer we have double-checked the manuscript and have corrected the mentioned errors.]

Reviewer 2 Report

Comments and Suggestions for Authors

Title:     

 Tocosomes as potent and novel nanonutraceutical delivery sys-

tems for health and biomedical applications

Authors:

Omar Atrooz , Elham Kerdari, M. R. Mozafari, Nasim Reihani, Ali Asadi, Sarabanou Torkaman,Mehran Alavi, Elham Taghavi

 This review aims to describe different aspects of tocosome, in parallel to liposome and nanoliposome technologies pertaining to nutraceutical and nanonutraceutical applications. Different properties of these nanocarriers, such as their physicochemical characteristics, preparation approaches, targeting mechanisms and their applications in the biomedical and nutraceutical industries are also covered. this article focus on the advantages of tocosomes, liposomes and nanoliposomes, with emphasis on their physicochemical properties, manufacturing techniques, targeting mechanisms, as well as their applications in the pharmaceutical, biomedical, food, feed and nutraceutical industries.

1.       Figure 1 : Some of the main advantages and benefits of using tocosome in the

biomedical and nutraceutical industries

Please add the references relative with the main advantages and benefits such as high absorption, prolonged release and others.

2.      Please add the references to more than 100 pieces.

Comments on the Quality of English Language

Minor editing of English language required.

Author Response

Comments 1: [Figure 1 : Some of the main advantages and benefits of using tocosome in the

biomedical and nutraceutical industries. Please add the references relative with the main advantages and benefits such as high absorption, prolonged release and others.]

Response 1: [Authors would like to thank this reviewer for instructive and useful comments and suggestions. New References are added to the legend of Figure 1 as suggested.]

Comments 2: [Please add the references to more than 100 pieces.]

Response 2: [We would like to thank this Reviewer for his time and valuable comments. A total of 12 new References are added to the manuscript - increasing Reference number to 84 . Although we were not able to increase the total number of References to 100, as suggested, the newly added References are mainly from 2023 and 2024 and they are completely relevant to the specific scope of the manuscript.]  

Round 2

Reviewer 1 Report

Comments and Suggestions for Authors

The authors appropriately responded to all comments. The manuscript was well revised and can be accepted.